# Transition metal dichalcogenide metamaterials with atomic precision

Battulga Munkhbat [1], Andrew B. Yankovich [1], Denis G. Baranov [1,2], Ruggero Verre[1], Eva Olsson [1] & Timur O. Shegai [1✉]

The ability to extract materials just a few atoms thick has led to the discoveries of graphene, monolayer transition metal dichalcogenides (TMDs), and other important two-dimensional materials. The next step in promoting the understanding and utility of flatland physics is to study the one-dimensional edges of these two-dimensional materials as well as to control the edge-plane ratio. Edges typically exhibit properties that are unique and distinctly different from those of planes and bulk. Thus, controlling the edges would allow the design of materials with combined edge-plane-bulk characteristics and tailored properties, that is, TMD meta-materials. However, the enabling technology to explore such metamaterials with high precision has not yet been developed. Here we report a facile and controllable anisotropic wet etching method that allows scalable fabrication of TMD metamaterials with atomic precision. We show that TMDs can be etched along certain crystallographic axes, such that the obtained edges are nearly atomically sharp and exclusively zigzag-terminated. This results in hexagonal nanostructures of predefined order and complexity, including few-nanometer-thin nanoribbons and nanojunctions. Thus, this method enables future studies of a broad range of TMD metamaterials through atomically precise control of the structure.

[1] Department of Physics, Chalmers University of Technology, 412 96 Gothenburg, Sweden. [2] Center for Photonics and 2D Materials, Moscow Institute of Physics and Technology, Dolgoprudny 141700, Russia. ✉email: timurs@chalmers.se

TMD materials have recently attracted a boom of scientific and technological activities due to the discovery of the direct bandgap of monolayer MoS$_2$ and its related exciton physics[1,2]. In addition to the excitonic aspect, these materials possess a number of attractive characteristics, including excellent mechanical[3], electronic[4,5], and catalytic[6–9] properties. Moreover, the inherent ability of two-dimensional (2D) materials to form van der Waals (vdW) heterostructures opens wide possibilities for designing hybrid systems with tailored properties[10]. Heterostructure physics is further enriched by intriguing Moiré patterns whose long-range order enables novel functionalities[11].

One-dimensional edges are another inherent property of any 2D material. Thanks to their reduced dimensionality, edges exhibit unique properties that are distinctly different from those of planes and bulk. Examples include metallic and ferromagnetic nature of zigzag-terminated edges and conversely semiconducting and nonmagnetic behavior of armchair edges as recently predicted and experimentally verified by density functional theory (DFT) and scanning tunneling microscopy studies[12–16]. Metallic edges exhibit high electro- and photo-catalytic activity[17], notably as non-precious analogs of platinum for the hydrogen evolution reaction[6–8]. Due to symmetry breaking at the edges, they also show potential for nonlinear optics applications, such as enhanced second harmonic generation[18]. Thus, controlling edges opens exciting possibilities in designing and exploring materials with synthetic and synergetic edge–plane–bulk characteristics, that is, TMD metamaterials. This approach is central to this work.

The problem of edge engineering in various vdW materials and their monolayers has been approached by many different methods, including chemical[19,20], surface engineering[21], thermal annealing[22,23], plasma treatment[24–26], and other emerging techniques[27]. However, many of these methods suffer from limitations, such as complexity, damage of excitonic properties, lack of precision, and harsh experimental conditions. More specifically, several of the most relevant techniques use a combination of top-down fabrication with plasma etching, similar to this work[24–26]. The first two works report methods developed specifically for anisotropic etching of mono- and few-layer graphene, while its applicability to TMDs remains to be verified. Etching of MoS$_2$ by highly reactive XeF$_2$ in gas phase was also previously reported[26]. To achieve anisotropic etching, however, this method requires a graphene mask, which adds complexity. Thus, despite significant progress in engineering edge states of vdW materials, the possibility to deterministically and reproducibly control the edges and edge–plane ratio of TMD materials remains relatively limited.

Here, we present a method that is free from the above-mentioned limitations and that allows to accurately control the edges and edge–plane ratio of various TMDs at nearly the atomically sharp limit. The process combines standard top-down lithography nanofabrication methods with subsequent anisotropic wet etching. This anisotropic wet etching process operates in a wide parameter range, from bulk crystals down to bilayers, and allows fabricating nanostructures with various sizes and shapes ranging from a single isolated hexagonal nanohole to densely packed arrays of nanoholes with lateral edge-to-edge separations as narrow as ~3 nm. Notably, the method operates at ambient conditions, over large areas, and uses only abundant chemicals, such as water solutions of hydrogen peroxide (H$_2$O$_2$) and ammonia (NH$_4$OH). Therefore, a variety of unexplored TMD nanostructures and metamaterials, such as semiconducting-plane metallic-edge composites and ultrathin nanoribbons, can be fabricated in a highly controllable and scalable fashion. Thus, this method opens possibilities for tailoring TMD material properties through extremely precise control of the structure.

## Results

**Summary of the fabrication process.** The three-step fabrication process is summarized in Fig. 1a (additional details are shown in Supplementary Fig. 1). In step (i), a TMD flake, such as for example tungsten disulfide (WS$_2$), of a certain thickness was isolated from a purchased crystal onto a thermally oxidized silicon substrate using standard mechanical exfoliation[28]. In step (ii), the exfoliated flake was nanopatterned using conventional top-down methods, such as a combination of electron beam lithography (EBL) and reactive ion etching (RIE), or by a focused ion beam (FIB) (see "Methods" for further details of FIB experiments). These kind of patterning techniques have been previously used to obtain various nanophotonic and excitonic structures[29,30]. However, these previously studied nanostructures were, to our knowledge, not well-defined in terms of edges, which likely consisted of an uncontrolled mix of zigzag, armchair and, possibly, amorphous configurations. To achieve atomically sharp edges with well-defined terminations, the flake that was pre-patterned in step (ii) was further exposed to an aqueous solution of H$_2$O$_2$ and NH$_4$OH (1:1:10 H$_2$O$_2$:NH$_4$OH:H$_2$O volume ratio of stock solutions) at mild heating ($T = 50$ °C) in step (iii). This exposure results in highly anisotropic wet etching of the pre-patterned circular holes in the flake, with etching occurring preferentially in-plane (Fig. 1a).

As seen from optical and scanning electron microscopy (SEM) images (Fig. 1b and Supplementary Fig. 2), an initial single circular hole with a radius of ~1 μm is transformed into a perfect hexagonal hole within ~8 min. The etching time required for formation of perfect hexagonal holes is determined by temperature, concentration and composition of the etching solution (Supplementary Figs. 2 and 3). For instance, by using an etchant solution composed of only water solution of hydrogen peroxide (H$_2$O$_2$:H$_2$O 1:10 volume ratio), the initial circular holes can also be etched into hexagonal holes, but at a slower etching rate (Supplementary Fig. 3). Note also that circular holes seen in the background of Fig. 1b (rightmost image) are due to the patterned SiO$_2$/Si substrate, which is left unetched by hydrogen peroxide. These holes are a result of overetching WS$_2$ flake by RIE in step (ii). Notably, the quality of the SiO$_2$ edges are limited by EBL and RIE, however, after wet etching all holes in WS$_2$ become extremely sharp (much sharper than the original holes obtained through EBL/RIE methods), regardless of the quality of the original hole.

A 70° tilted SEM image shows that the obtained hexagonal holes have sharp sidewalls and are all oriented in the same way (Fig. 1c). This orientation follows the hexagonal crystallographic symmetry of WS$_2$ crystal along [0001], as we show below. The hexagonal shape and extreme sharpness of the obtained structures imply that the etching mechanism is self-limited by certain crystallographic planes, similar to the well-known anisotropic wet etching of single crystalline silicon[31].

**High-resolution electron microscopy.** To verify the sharpness and terminations of the obtained edges, we performed a high-resolution transmission electron microscopy (HRTEM) study, which shows that the etched sidewalls are nearly atomically sharp with surface roughness variations of a few atomic planes (Fig. 1d, e and Supplementary Fig. 4). Selected area electron diffraction reveals the orientation of the etched hexagons are always synced to the TMD crystallography such that the etched surfaces are perpendicular to the <1100> direction (inset Fig. 1d). This confirms that the etched surfaces are exclusively aligned with the zigzag-terminated direction, indicating the edges are likely zigzag terminated. This observation agrees with earlier DFT results,

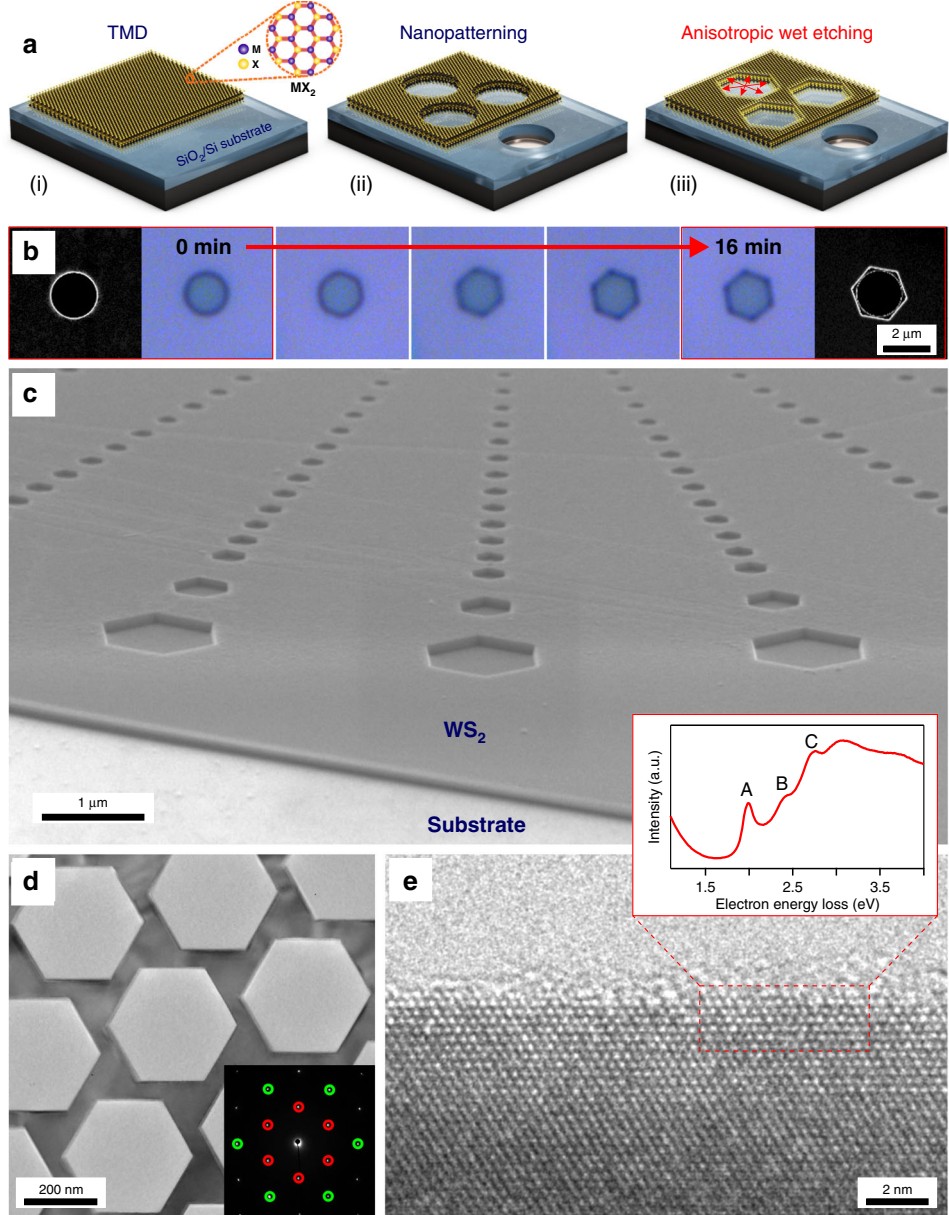

**Fig. 1 Etching hexagonal nanostructures in TMD materials. a** Schematic of the fabrication method: (i) transfer of a mechanically exfoliated TMD flake on a $SiO_2$/Si substrate, (ii) nanopatterning of the transferred flake using standard top-down lithography, and (iii) anisotropic wet etching of the pre-patterned flake by water solution of hydrogen peroxide and ammonia. **b** Time-resolved anisotropic wet etching for a single hole in a $WS_2$ multilayer flake. Leftmost and rightmost panels show SEM images of the hole after 0 min and 16 min of etching, respectively. **c** 70° tilted SEM image of anisotropically etched $WS_2$ flake, revealing hexagonal holes and their orientation. **d** TEM image of an etched $WS_2$ hexagonal array. The inset shows a selected area electron diffraction pattern from the imaged area, revealing a single crystalline $WS_2$ diffraction pattern along a [0001] zone axis with an orientation that confirms the etched surfaces are aligned with the zigzag-terminated direction. The red and green circles identify the 1100 and 1120 families of diffraction spots, respectively. **e** HRTEM image of an etched surface, revealing its nearly atomically sharp zigzag nature. The inset shows a scanning TEM (STEM) EELS spectrum acquired with the electron probe positioned directly at the etched zigzag surface, revealing the presence of strong A-, B-, and C-exciton signals.

which predict that zigzag edges are more stable than armchair edges[13,32].

Low-loss electron energy loss (EEL) spectra acquired from the etched surfaces show strong signals from the A-, B-, and C-excitons (inset in Fig. 1e and Supplementary Fig. 5), confirming that the $WS_2$ material is of high quality and the excitonic properties are undamaged by the etching process to the very edge of the material. Low-loss EEL spectra from the etched surfaces also reveal the presence of a signal at ~13.8 eV (Supplementary Fig. 6) that is similar to a signal which has been previously

associated with zigzag edges in $MoS_2$[33]. TEM images reveal that a thin layer (0–3 nm) of amorphous material occasionally appears on top of the etched zigzag surfaces (Fig. 1e and Supplementary Fig. 4). This material is likely due to left-over residues originating either from the etching process or from the scotch-tape and polymer stamps used during the dry-transfer process for TEM sample preparation (see "Methods"). Additional data about the quality of the edges, including spectroscopic analysis of the elemental composition and optical signals are provided in the Supplementary Information (Supplementary Figs. 4–8).

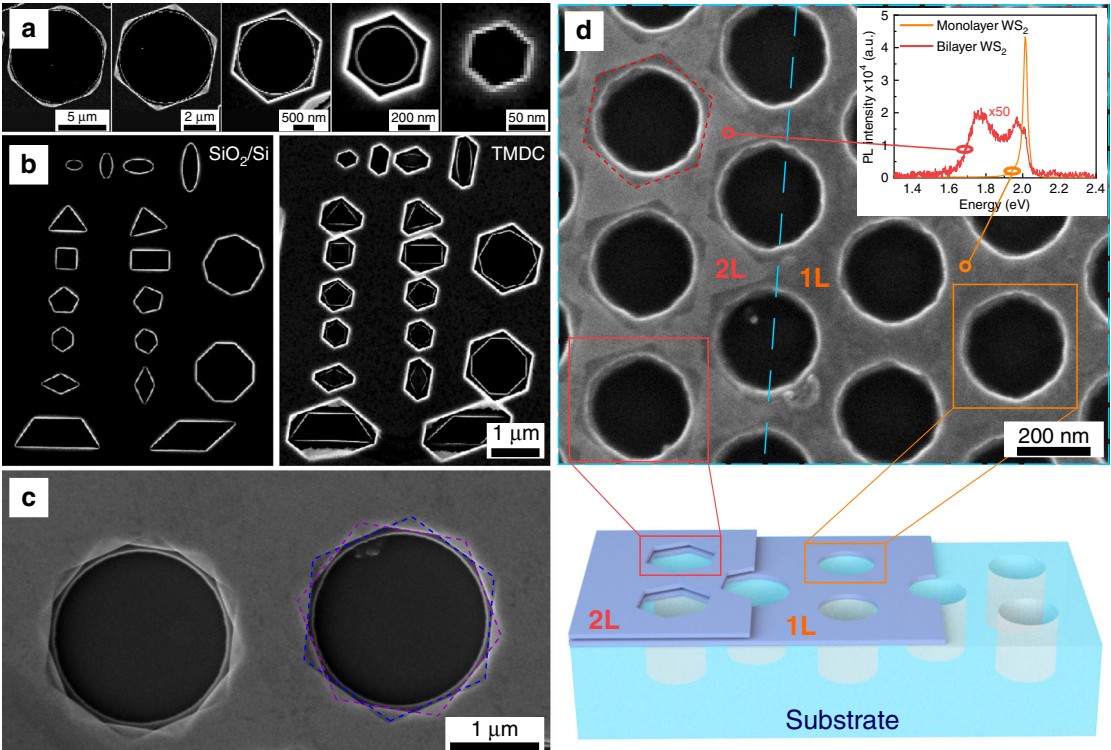

**Fig. 2 Etching individual nanoholes. a** Effect of hole size: SEM images of etched holes in WS$_2$ with various sizes from ~50 nm to ~10 μm in diameter. **b** Effect of hole shape: SEM images of single holes with different initial shapes, such as pentagons, octagons, and ellipsoids, etched into both a bare SiO$_2$/Si substrate and a WS$_2$ flake, illustrating the comparison between initial and anisotropically etched holes. **c** Effect of heterostructure: SEM image of etched hexagonal holes in a WS$_2$ multilayer heterostructure that has a 30° rotation between the stacked flakes. **d** Effect of flake thickness: an SEM image and a corresponding schematic sketch showing formation of hexagonal holes in a bilayer and circular holes in monolayer WS$_2$. The inset shows the photoluminescence spectra of bi- (red curve) and mono-layer (orange curve) WS$_2$.

**Etching individual nanoholes**. To explore the morphological diversity of the proposed etching method, we first investigated the behavior of single isolated nanoholes. In particular, these studies focused on the effect of hole size and shape (Fig. 2a, b, respectively), heterostructure (Fig. 2c) and TMD thickness (Fig. 2d) on the resulting etched structure.

Figure 2a and Supplementary Fig. 9 show that it is possible to etch a WS$_2$ multilayer to produce hexagonal etched holes with diameters varying in a broad range—from 50 to 10,000 nm (tested here, however, there seems to be no upper limit on the hole size). It is also important to mention that small circular holes rapidly transform into perfect hexagonal holes, whose subsequent etching is almost entirely stopped (see Supplementary Fig. 2). Larger holes, however, converge to perfect hexagons at a much slower pace. Thus, if the etching is stopped before the perfect hexagonal shape is reached, the resulting hexagonal holes appear incomplete.

In addition to the effect of the hole size, we have examined the effect of hole shapes (Fig. 2b). A variety of noncircular shapes, including ellipses, triangles, squares, pentagons, and trapezoids were tested and all of them were converted into regular or irregular hexagonal shapes, irrespectively of the shape and size of the original hole (Fig. 2b). This suggests that random defects on a TMD surface can be anisotropically etched and converted into zigzag-terminated edges. Additional data about the quality of single holes and more complicated geometries such as bull's eye nanostructures, as well as tests performed on multiple semiconductor group VI TMD materials, including MoS$_2$ (and its mineral form), MoSe$_2$ and WSe$_2$, are given in the Supplementary Information (Supplementary Figs. 10 and 11).

We further investigated a single hole in a TMD heterostructure (Fig. 2c), which, as it turns out, can also be etched into a hexagon.

Such anisotropic etching is possible despite the fact that the crystallographic axes between different layers within a heterostructure do not, in general, coincide with each other. Etching a two flake heterostructure where the top flake (WS$_2$) was rotated by ~30° with respect to the bottom flake (another WS$_2$) resulted in hexagons rotated by ~30° (Fig. 2c and Supplementary Fig. 12). This confirms the etching is anisotropic even for heterostructures and suggests that this method can be used to reveal crystallographic orientation of different flakes with respect to each other.

To verify the influence of the flake thickness on the etching process, we have investigated several different thicknesses, ranging from a few hundred nanometers down to a monolayer. Here, it is important to emphasize that monolayer TMDs have recently attracted substantial attention due to their direct bandgap and interesting excitonic physics[1,2]. As shown in Fig. 2d and Supplementary Fig. 13, a WS$_2$ monolayer withstands the etching process, as is evident from a bright photoluminescence (PL) signal, originating from the A-exciton. This observation ensures the excitonic properties of the monolayer are preserved upon interaction with hydrogen peroxide. However, the originally fabricated circular nanoholes in a monolayer cannot be directly patterned into hexagonal shapes; instead they transform into irregular triangular shapes (Supplementary Fig. 13b–e). Further details about anisotropic etching of monolayers are discussed in Supplementary Notes (section d) and in Supplementary Fig. 13. In contrast to the monolayer behavior, an adjacent WS$_2$ bilayer is etched into hexagonal shapes just like any other multilayer, as shown by SEM (Fig. 2d). The bilayer nature is confirmed by the PL signal (inset Fig. 2d). This result has three important implications. First, it confirms that the etching mechanism has an anisotropic nature and does not etch along the basal plane

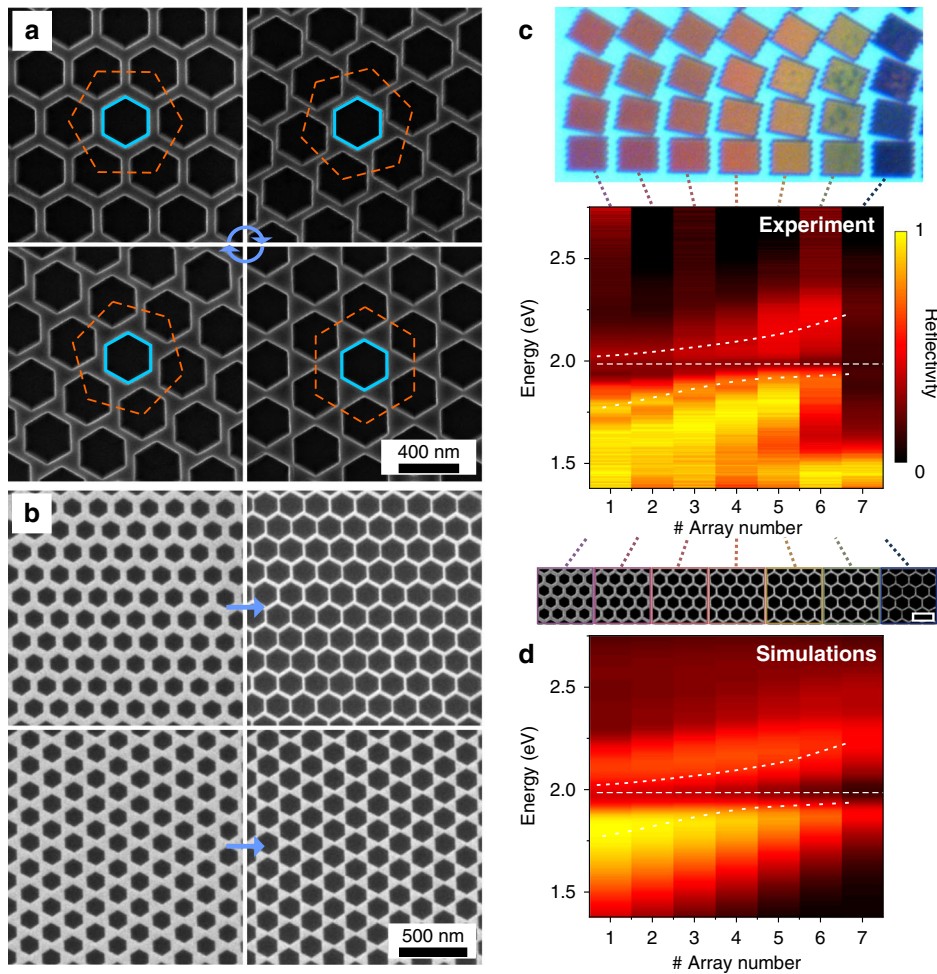

**Fig. 3 A variety of etched hexagonal hole arrays and their optical properties. a** SEM images of hexagonal hole arrays with various rotational angles (15° step) with respect to the orientation of individual hexagonal holes in a WS$_2$ flake. The orange-dashed and blue hexagons identify the orientations of hexagonal lattice arrays and individual hexagonal holes, respectively. **b** SEM images of honeycomb and bow-tie hexagonal hole arrays with fixed lattice periodicity but different hole sizes (bigger holes are shown to the right). **c** True-color optical microscope image and reflectivity spectra of hexagonal arrays with various hole sizes and lattice orientations in ~70 nm thick WS$_2$. The bottom row of the optical image consists of honeycomb arrays with variable hole sizes ranging from 240 to 300 nm (left to right) and a fixed lattice pitch of 400 nm, and their SEM images are shown, respectively. The scale bar is 500 nm. **c** Experimental and **d** simulated reflectivity spectra. Dashed lines are guides for an eye, indicating the A-exciton energy and the avoided crossing.

[0001], since the bilayer remains the bilayer (and does not get etched into a monolayer or further) throughout the entire etching process. Second, it suggests that mono- and multi-layers not only have different optical properties, but they might also have different chemical properties. We note that despite the impossibility of directly etching monolayers into hexagonal structures using our method, etched monolayers could potentially be fabricated by exfoliating etched multilayer structures[28]. Third, the results above highlight the uniqueness of our anisotropic etching method, as it operates exclusively with multilayers, starting from bilayers and up to thicknesses of hundreds of nanometers (thicker samples were not tested in this study). This implies that the process of hexagonal etching inherently requires a multilayer nature of the TMD flake. We return to this point and the mechanism of anisotropic etching of multilayers in the Discussion section.

**Etching nanohole arrays**. To explore more complicated and extended structures, as well as to test the scalability of our approach, we fabricated regular arrays of hexagonal nanoholes (Fig. 3). Our results show that the arrays' geometry can be

accurately and deterministically controlled over the entire flake dimensions (up to several hundreds of microns and is limited by the flake size), thus forming an extended TMD metamaterial with nearly atomic precision in the structure. Moreover, the pattern quality is limited not by the size of the flake itself, but rather by the methods used to pre-pattern the sample in step (ii), that is, by EBL and RIE (Fig. 3 and Supplementary Figs. 14–16). Various periodic patterns, such as honeycombs, vortexes, and bow-tie arrays, can be fabricated by controlling the orientation of the array with respect to the crystallographic axis of the WS$_2$ material (Fig. 3a and Supplementary Fig. 16). The orange-dashed and blue hexagons in Fig. 3a outline the orientations of hexagonal lattice and individual hexagonal holes, respectively. The precise control over the parameters of the periodic lattice with respect to the given orientation of individual hexagonal holes (which follows the crystallographic orientation of the WS$_2$ flake) enables fabrication of a rich variety of nanostructured arrays. The sidewalls in these arrays are as sharp as the single nanohole structures shown previously, suggesting the method is scalable. Furthermore, because the etching time and the initial circular hole size and location can be accurately controlled, a variety of complex TMD nanostructures with preferential zigzag edges can be readily

fabricated over large areas (Fig. 3b). Importantly, many of these nanostructured arrays are challenging to obtain using alternative methods, whereas the method described here combines a high level of versatility, precision, and control with scalability over macroscopic dimensions.

True-color optical image of the obtained hexagonal lattices in ~70 nm thick $WS_2$ (Fig. 3c) reveals bright structural colors, which are hole size and arrangement dependent. This suggests that nanopatterned $WS_2$ arrays act as photonic crystals, making them interesting for structural color applications and tailored light–matter interactions. Importantly, many TMD materials such as $MoS_2$ and $WS_2$ have exceptionally high values of the in-plane refractive index $n > 4$ and optical anisotropy[29,34–36], which, together with the etching method discussed here, enrich the TMD nanophotonics toolbox. The unpatterned multilayer $WS_2$ flake (thickness ≈ 70 nm) appears blue in the background due to a peculiar self-hybridized interference[35] (reflectivity spectrum is shown in Supplementary Fig. 18).

The structures shown in the true-color optical image (Fig. 3c) are organized such that the arrays displaced vertically vary in orientation with respect to the $WS_2$ crystallographic axis, whereas the arrays displaced horizontally vary in the hole size. The array orientation is varied in 10° steps, resulting in a transition from honeycomb (bottom row) to bow-tie shaped (top row) structures. The hole diameters range from 240 nm (far left) to 300 nm (far right) with a step of 10 nm and a fixed lattice pitch of 400 nm. The honeycomb array data (experimental and simulated reflectivity spectra, as well as SEM images) are shown in Fig. 3c, d. The reflection spectra reveal interesting resonant features, which linearly disperse with the hole size in the near-infra red region (<1.9 eV). The spectra show additional avoided crossing around the $WS_2$ A-exciton (~2 eV), which can be potentially attributed to hybridization between optical modes of the array and $WS_2$ exciton, similar to a previous study on individual $WS_2$ nanodisks[29], however, a rigorous eigenmode analysis is necessary to confirm the hybridization.

To elaborate more on the nature of these optical resonant features, we performed finite-difference time-domain (FDTD) simulations (see "Methods"), which confirm the avoided-crossing behavior, as well as qualitatively reproduce the essential spectroscopic features of experiments (Fig. 3d). Additional FDTD data, including transmission and absorption plots for all studied arrays are shown in Supplementary Fig. 18. Note that array #7 exhibits a weak reflectivity (~15%) over the entire visible spectrum and a dark-blue color. This result suggests that this specific array, together with the non-transparent $SiO_2$/Si substrate below, is highly absorptive, which is corroborated by FDTD results (Supplementary Fig. 18). Interestingly, SEM data show that this specific array is the most depleted in terms of the amount of $WS_2$ material left after etching and nanopatterning (Fig. 3d).

In closing this section, we stress that nanostructured $WS_2$ arrays studied in Fig. 3 combine optical resonant features, semiconducting and excitonic properties, large surface-to-volume ratio, and exclusive zigzag edges. Such combination offers prospects for cross-fertilization between various functionalities within these structures.

**Ultranarrow nanostructures**. A combination of accurate control over the etching rate and sharpness of the etched zigzag edges enables fabrication of various ultranarrow nanostructures (Fig. 4). The top row in Fig. 4 shows nanoribbons with remarkably sharp vertical walls in densely packed arrays. It has been theoretically predicted that ultrathin TMD nanoribbons have drastically different optical, excitonic, and electronic properties compared to conventional TMD materials[12,32]. However, experimental

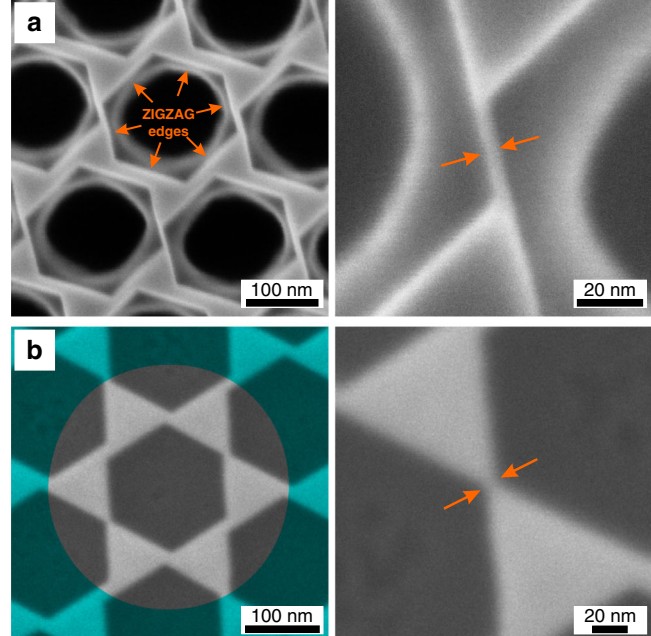

**Fig. 4 Fabrication of a few-nanometer narrow nanoribbons and nanojunctions. a** SEM images of a (left-panel) hexagonal hole array in $WS_2$ and a (right-panel) thin (~3 nm) nanoribbon sandwiched between two zigzag-terminated edges. **b** SEM images of a (left-panel) hexagonal hole array in $WS_2$ resulting in bow-tie nanostructures and a (right-panel) thin nanojunction structure.

evidence for this is lacking because of the difficulty in fabricating these nanostructures. Furthermore, the bow-tie array etched down to a sharp vertical wall limit results in narrow nanojunctions (Fig. 4b). The obtained nanoribbons and nanojunctions can have edge-to-edge distances of about ~3 nm. Importantly, producing even smaller nanoribbons and nanojunctions could be possible since the currently obtained size features are limited by the quality of the initial patterns created by the conventional top-down nanofabrication methods. The sizes of these nanoribbons and nanojunctions are compact enough to approach the size of the exciton Bohr radius in $WS_2$ multilayers, which could lead to additional quantum confinement effects. Furthermore, these structures have high surface-to-volume ratios, which could be useful for catalytic[6–8] and nonlinear optics[18] applications.

## Discussion

The appearance of ultranarrow features displayed in Fig. 4, together with the data presented earlier, points towards the mechanism of the anisotropic wet etching. Specifically, in a TMD of a general formula $MX_2$ there exist many types of edge states, including both X- and M-terminated zigzag edges. The hypothesis is that these different zigzag and armchair edges are likely different in terms of their chemical stability against etching by hydrogen peroxide (this hypothesis is supported by earlier literature studying the opposite behavior of S- and Mo-edges in chemical vapor deposition grown $MoS_2$ clusters[12]). Thus, etching of a $MX_2$ monolayer likely results in survival of only one type of the zigzag edges, whichever is more stable (the armchair configuration does not withstand the wet etching process). This is partially confirmed in Supplementary Fig. 13, which shows that anisotropic etching of a monolayer results in an equilateral triangle, instead of a hexagon (note that the structure of an equilateral triangle requires that all edges within the structure have the same X- or M-zigzag termination). In 2H-stacked multilayers, which most common TMDs follow, the X- and M-terminated

zigzag edges within a hexagon overlap each other in a vertical stack. This vertical sequencing likely makes the hexagonal arrangement ultimately possible in a multilayer TMD configuration. We note, however, that future studies are required to confirm this hypothesis and the exact mechanism of anisotropic etching process studied here is likely more complex.

Regardless of the exact etching mechanism, it is important to recognize another important consequence of the reported process in relation to the formation of ultranarrow nanoribbons and nanojunctions (Fig. 4). It is only possible to obtain these structures starting from a multilayer sample. Indeed, since the monolayer case supports the survival of only one type of zigzag edges, the final structure cannot result in a monolayer nanoribbon. Otherwise its opposite ends would be terminated by different atoms, which is prohibited by variability in M- and X-terminated zigzag edges stability hypothesis. Thus, our method allows fabricating unique quantum confined structures starting from TMD multilayers (Fig. 4).

In summary, we have demonstrated an anisotropic wet etching method that enables fabrication of complex hexagonal nanostructures with nearly atomically sharp zigzag edges in various TMD materials ($WS_2$, $MoS_2$, and $MoSe_2$ successfully tested here) using only abundant and cheap chemicals such as hydrogen peroxide and ammonia at ambient conditions. Our method allows fabricating individual nanostructures, complex nanostructured arrays, nanopatterned heterostructures, and vertically stacked ultrathin TMD nanoribbons, which to the best of our knowledge is not possible using other methods. We unite all these structures under a common umbrella of TMD metamaterials, possessing intermixed plane-edge characteristics, whose detailed material and optical properties remain to be explored.

We envision that the combination of high surface-to-volume ratio and exclusively zigzag edges, which arise in these TMD metamaterials, could be useful for multiple purposes, including TMD-based catalysis[6,7] and sensing[37] applications. Furthermore, there is a potential for other areas and applications, such as photodetectors[38], quantum transport[4], nonlinear optics[18], nanophotonics[29], and optomechanics[39] to benefit from the method. Finally, we speculate that it may be possible to generalize a modified version of this method to other 2D material classes, such as graphene, hexagonal boron nitride, Mxenes, and others, because the anisotropic etching mechanism is likely related to different relative stabilities of various edges and planes (zigzag, armchair, and basal), which is a generic property of any vdW material with hexagonal crystallographic symmetry.

## Methods

**Sample fabrication.** *Exfoliation*: TMD flakes were mechanically exfoliated from bulk crystals (HQ-graphene or mineral crystal) onto polydimethylsiloxane (PDMS) stamps using the scotch-tape method, and then transferred onto various substrates using the all-dry-transfer method[28]. Flakes with different thicknesses were transferred on different substrates[28] and the thickness was measured using a VEECO profilometer. The initial patterns of the TMD flakes were performed by either a combination of e-beam exposure of a positive resist and dry etching, or direct FIB milling.

*EBL*: Nanopatterning the TMD flakes with e-beam lithography was carried out by first spin coating a positive ARP 6200.13 resist at 6000 rpm for 1 min, which yields ~400 nm thick resist. Samples were exposed using a JEOL JBX 9300FS EBL system operated at 100 kV. The alignment of the flakes was ensured using a pre-fabricated marking system. The samples were then developed using n-Amyl acetate for 4 min and dried with nitrogen gas. Subsequently, the samples were dry etched in an inductively coupled plasma/reactive ion etcher (RIE) by $CHF_3$ (50 ccm) and Ar (40 ccm) at a pressure of 10 mTorr and accelerated by 50 W forward power. An etching rate of ~10 nm per minute was measured for $WS_2$ and the dry etching time was modified accordingly for the different sample thicknesses. During the dry etching process, the top resist layer may get hardened, which leads to an issue of removing the left-over resist. Therefore, oxygen plasma stripping was performed with $O_2$ (40 sccm), after the dry etching process, to remove not only the hardened resist layer, but also substantial amount of the resist. Subsequently, the left-over resist-mask was removed by acetone at 50 °C for 3 min and rinsed with isopropyl alcohol and deionized water.

*FIB*: Nanopatterning TMD flakes with FIB milling was carried out using a Tescan GAIA3 FIB SEM instrument. The ion milling procedure was performed with a low energy electron beam of 40 kV and an ion beam current of a few tens of pA (see Supplementary Fig. 17).

*Anisotropic wet etching*: To perform the preferential wet-chemical etching, the patterned samples were immersed in an aqueous etchant solution composed of $H_2O_2$:$NH_4OH$:$H_2O$ with a volume ratio of 1:1:10, or $H_2O_2$:$H_2O$ with a volume ratio of 1:10 at 50 °C. The volume ratio is referred to stock solutions. The stock solutions were: $H_2O_2$—31% in water and $NH_4OH$—25% in water, respectively. The etching rate can be precisely controlled and optimized by varying the temperature, the etching time, the concentration, and the composition of the etchant solutions. Mild heating of 50 °C allows us to slow down the etching rate to precisely control the final size of ultranarrow structures. Otherwise, a combination of higher concentration of etchant with higher temperature can be employed to speed up the etching rate for larger structures. During the wet anisotropic etching process, the pre-patterned circular holes in the TMD materials are transformed into hexagonal-shaped holes that follow the TMD crystallography and have exclusively zigzag edges. After etching, the samples were rinsed in deionized water to get rid of chemical residues, dried with $N_2$ gas and then characterized with optical microscopy and SEM using a Zeiss ULTRA 55 FEG and a JEOL 7800F Prime.

**TEM and STEM EELS characterization.** The samples for TEM imaging and STEM EELS experiments were prepared by first nanopatterning and etching the $WS_2$ flakes on regular $SiO_2$/Si substrates, as mentioned above. The pre-patterned and anisotropically etched samples were mechanically exfoliated onto a PDMS stamp using scotch-tape. Then the exfoliated flakes were transferred from the PDMS stamp onto a TEM grid composed of nine ~10 nm thick SiN TEM membrane windows (TEMwindows.com). Some of the membranes were purposefully broken to enable TEM observation of the flakes suspended over vacuum with no substrate.

S/TEM experiments were performed on a JEOL Mono NEO ARM 200F that is equipped with a Schottky field emission gun, double Wien filter monochromator, probe aberration corrector, image aberration corrector, and Gatan Imaging Filter continuum HR spectrometer. Experiments were conducted at 60 keV, which is below the electron beam induced knock-on damage threshold for $WS_2$. STEM EELS spectrum images were acquired using a probe current of ~180 pA and a probe size of a few Angstroms. Dual EELS mode was used to enable post-acquisition energy instability correction and high signal-to-noise ratio of the desired low-loss or core-loss signals. This was accomplished by using the first spectrum image for only capturing the zero-loss peak region with a very short pixel dwell time. The second spectrum image was used for capturing the low-loss or core-loss regions with the zero-loss peak shifted off the detector to enable longer pixel dwell times and improved signal-to-noise ratio. Energy-dispersive X-ray (EDS) spectrum images were acquired simultaneously with the dual EELS spectrum images, which confirm the elemental distributions observed in EELS. However, we only show the EELS data because they exhibit improved signal-to-noise ratio and are better for light-element analysis with these samples. EELS signal maps were created by integrating the signal of the identified EELS core-loss edge after a background model was subtracted from each spectra. The background model was determined by fitting the pre-edge region to a power law.

**Optical characterization.** Bright-field reflectivity spectra at normal incidence were collected using a 20× air objective (NIKON, NA = 0.45), directed to a fiber-coupled spectrometer and normalized with reflection from a standard dielectric-coated silver mirror.

**FDTD simulations.** Numerical simulations of the electromagnetic response of nanopatterned $WS_2$ arrays were performed with the FDTD method in commercial software (Lumerical, Canada). $WS_2$ was described by an anisotropic permittivity tensor. The in-plane permittivity component was approximated by data from[40] for bulk $WS_2$; the out-of-plane component was approximated by a constant value of 6.25 in accordance with previous simulations of $WS_2$ nanostructures[29]. Transmission and reflection spectra were obtained using a normal incidence linearly polarized plane wave source, in-plane periodic boundary conditions and steep angle perfectly matched layer boundaries in z-direction. The mesh accuracy parameter was set to 5.

## Data availability

The experimental and calculated optical spectra, SEM and TEM images, and EELS data that support the findings of this study are available through figshare.com with the identifier(s) https://doi.org/10.6084/m9.figshare.12786314.v1. Additional data are provided in Supplementary Information and are available from the corresponding author upon reasonable request.

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

## Acknowledgements

B.M., D.G.B., and T.O.S. acknowledge financial support from the Swedish research council (VR Miljö Grant No. 2016-06059). B.M. thank Prof. S. Davaasuren and Dr. G. Munkhbayar for providing a test mineral $MoS_2$ and fruitful discussions. FDTD simulations were supported by the Russian Science Foundation (grant 19-79-00362). A.B.Y. and E.O. acknowledge financial support from the European Union's Horizon 2020 research and innovation program under grant agreement No. 823717-ESTEEM3 and from the Swedish Research Council (VR) under Grant No. 2016-04618. This work was performed in part at the Chalmers Material Analysis Laboratory, CMAL.

## Author contributions

B.M. and T.O.S. conceived the general approach. B.M. fabricated samples and performed optical and SEM experiments. R.V. contributed to nanofabrication. A.B.Y. performed TEM and EELS experiments. A.B.Y. and E.O. analyzed TEM and EELS data. D.G.B. performed numerical FDTD simulations. All authors contributed to writing the manuscript. E.O. and T.O.S. supervised the project.

## Funding

## Competing interests

The authors declare no competing interests.
