## [Peer Review File · Nature Communications]

REVIEWER COMMENTS

Reviewer #1 (Remarks to the Author):

This work reported a scalable method to fabricate hexagonally shaped TMD nanostructures with smooth edges, by combining beam-based nanolithography like EBL and FIB with anisotropic chemical etching. The demonstrated several applications of the technique. First, they demonstrated formation of ~ 3 nm wide WS₂ nanoribbons and nanojunctions, which are otherwise difficult to fabricate using existing fabrication techniques. Second they demonstrated honeycomb-array WS₂ metamaterials with tailored optical properties. The method is simple and could be extended to other 2D materials. The work is well executed with several innovations which should appeal to the 2D materials and metamaterials communities. However, the authors to not provide enough detail to reproduce the work, and the introduction is disorganized and does not do a good job motivating the work or addressing the most relevant previous works. After addressing these issues, I will recommend this paper for publication.

Most important:

1. The authors did not provide nearly any details about the key innovation in their work: the anisotropic chemical etching process, which prevents the work from being reproduced. They need to clarify key etch parameters in the main text, such as the composition and concentration of the etchant, etch temperature, etch rate, and etch time, and any peripheral processing steps needed to make it work. They also need to describe how they choose the right chemical etchant.
2. The authors did not do a good job articulating what distinguishes their work compared to the most relevant examples literature. For instance, the paper *Advanced Materials* 2011, 23, 3061–3065 (Reference 24) demonstrated fabrication of nearly identical nanostructures in graphene by combining EBL with anisotropic plasma etch. Another paper *Nano Research* 2013, 6(3): 200–207 (Not referenced) reported anisotropic etching of MoS₂, generating hexagonal patterns. This reviewer believes that this paper has novelty compared with those previous works, but it is important that the authors articulate how their work is distinct with a direct comparison to the two papers mentioned above, and to moderate their language about novelty of their own anisotropic etch process.

Minor comments:

1. The discussion and structures in Figure 2 are difficult to follow. Figure 2d and the difference between monolayer and bilayer behavior is particularly difficult to interpret. This figure would benefit from cross-sectional diagrams of the proposed structure including the masking layers, so it is clear which layers are etching.
2. The introduction is misleading and poorly organized. It spends a lot of time motivating general concepts of 2D materials, heterostructures and edge physics. This is misleading, because the paper does not demonstrate any property of edges or heterostructures. The introduction would be much improved by spending more time discussing the challenges and progress of etch control (including the papers mentioned above), then focusing on the progress and opportunities of the actual applications they focus on in the paper (e.g. metamaterials).

Reviewer #2 (Remarks to the Author):

The authors report on anisotropic wet etching effect in transition metal dichalcogenide (TMD). The obtained edges in as-produced hexagonal holes are extremely smooth and exclusively zigzag-terminated. They further extend their study to fabricate more complex geometry pattern array and sub-10nm nanostructures including nanoribbons and nanojunctions. These results establish a facile and controllable method to tailor TMD materials, with a high potential in TMD-based electronics. The data are thoroughly presented. I would like to recommend its publication after major revisions.

Several issues need to be clarified or revised.

A key point for this study is to use H₂O₂/NH₄OH solution replacing other reported etchants. However, the author described nothing about this issue in the text. I was totally puzzled until I checked supplementary materials. This is not advisable for a research article. The authors should provide a recipe regarding the composition and proportion. I would further inquire the etching mechanism, the role of NH₄OH and origination for the advantage of this etching solution as compared to other etchants.

I doubt the claim of atomic precision. The author provided TEM data, showing edge roughness variation of 1-2 atomic plane in the selected area with tens of nanometer in length. Although it is not feasible to characterize the roughness in a large length scale as well as atomic precision, if the author could also provide roughness of 1-2 atomic plane for the triangle in the corner of hexagonal holes, the claim of atomic precision is then convincing.

There is almost no quantitative description about the relationship between etching speed, etching time and etching condition, which is much important for the authors to design and fabricate nanostructures.

The authors mentioned the etching in monolayer TMD is not anisotropic. Why? As I know, monolayer graphene on SiO₂ substrate also shows such isotropic etching (Adv. Mater. 23, 3061 (2011)), but on substrate with atomically smooth surface like hBN shows anisotropic etching again (Appl. Phys. Lett. 109, 053101 (2016)). The authors need to provide detailed discussion on this issue.

Reviewer #3 (Remarks to the Author):

The article “Transition metal dichalcogenide metamaterials with atomic precision” presents a fabrication technique to produce nanopatterned structures from WS₂ layers that can be further applied to a wider family of TMD materials, such as MoS₂ and MoSe₂. Moreover, the authors suggest that the developed fabrication technique can result in atomically sharp edges such as zigzag-terminated structures. Authors present successful fabrication using this technique of a wide range of interesting nanostructures, including particularly interesting results on heterostructures. In general, the control over the TMD material edge structure is of high importance for a wide range of applications, since such nanoscale precision patterning techniques give an opportunity to not only control electronic and optical properties of the materials but also create novel properties such as for example ferromagnetism. Therefore, the presented results are of high importance and interest to the wide reader audience of the Nature Communication journal. However, the current manuscript lacks important discussions such mechanism of chemical treatment, material quality preservation and photonic structure properties. Please see the major and minor correction list below. At this stage, the manuscript is not ready for publication, but it can be improved after the major corrections being addressed.

Major corrections:

- (1) WS₂ edge termination: The authors suggest that after the fabrication technique, the WS₂ structure termination has atomically sharp zigzag nature. However, in fig.1(e) it is very difficult to see the actual edge. The authors are strongly advised to include higher magnification image and atomic model of the material structure to justify the edge termination. It would be also very interesting to see the crystal corner similarly to the work by Chen, Q. et al. *Atomically Flat Zigzag Edges in Monolayer MoS₂ by Thermal Annealing. Nano Lett. 17, 5502-5507, (2017)*. In this work, Chen et al has also demonstrated that there are atomic corrugations present on the formed edge produced by thermal annealing, it would be very interesting to compare this result to the results of the proposed method.
- (2) Wet etching process: The presented fabrication technique relies fully on the anisotropic etching process activated by H₂O₂. However, the manuscript doesn't include any details on the etchant composition and the authors just refer to the solution as “etchant”, which is quite uninformative. Furthermore, there is no description of the mechanism of this etchant anisotropic functions. The authors are therefore strongly advised to include the chemical mechanism description. It might be very interesting to look into this, since it might further explain the results presented for single layer WS₂ where triangular structures are formed. This can be a very nice discussion section where all the results for various thicknesses and shapes are summarised and explained. Furthermore, it is also quite interesting why this etching technique works for MoS₂ and MoSe₂, but fails for WSe₂. Additionally, was the crystal thickness same for the tested structures of different materials? Based on the results presented for various shapes in Fig2, it's quite interesting to see when the self-termination happens, perhaps it might be possible to develop a model describing the self termination process similar to results presented in Sang, X. et al. *In situ edge engineering in two-dimensional transition metal dichalcogenides. Nat. Commun. 9, 2051, (2018)*.
- (3) Material quality preservation: Another very important question is how much damage is introduced during the etching process. While the PL spectrum shown in the Fig. 2d shows high quality WS₂ in the middle, it would be very interesting to see the line scan of PL or Raman for example to estimate how much material is damaged close to the edge of the structures. This becomes a very crucial question especially for a nanojunction structures presented in Fig. 4. Additionally, it would be very nice to see the PL comparison for the material before and after patterning, since this can give additional measure of the induced

defects, although the excitonic states will probably be modified too. From this point of view, having PL measurements for structured materials would be potentially quite interesting to study.

- (4) Photonic structure properties: The presented fabrication technique results in very interesting photonic structures as presented in Fig.3. It would be very nice to see the actual measured reflectance spectra for unstructured and structured material regions, as well as supporting substrate. Additionally, the reader would definitely benefit from small SEM images of each array to be able to quickly judge how much WS₂ material is actually left after patterning. For example, array N7 is claimed to have high absorption, however there is probably quite little WS₂ material left at least based on the structure sizes mentioned. Perhaps the observed reflectance results can be supported by numerical calculations, such as FDTD calculations, which should be quite reasonable for 70nm thick WS₂. While Fig.3a,b are quite nice demonstrations of the fabrication technique cases, it appears that these results can be either merged with Fig.2 or moved to the supplementary. The reader would probably benefit more from a more detailed figure on the photonic crystals. The paper might be stronger with an additional deeper discussion of the expected photonic structure properties, i.e. a theoretical part with calculations of the expected resonant features for the selected photonic crystal geometries. This becomes particularly relevant since the article title is *metamaterials*.
- (5) Nanoribbons: Fig. 4 shows very interesting results demonstrating nanoribbons and nanojunctions. However, all the presented measurements are done with SEM, which is known to locally modify WS₂ as well. AFM measurements would give more precise results on the nanostructure sizes, while not modifying the 2D materials. Additionally, Fig. 4a clearly shows an internal ring structure that is not discussed in the text, are those residues or there is an etching tilt or this is etching in SiO₂? Similar rings are observed in Fig.2 d and c.
- (6) FIB results are quite interesting especially from the point of view of polymer residue influence on the process. However, they are not discussed in the main text.

Minor corrections:

1. Additional data about the quality of single holes, more complicated structures such as bull's eye nanostructures, as well as tests performed on multiple semiconductor group VI TMD materials ~~is~~ **are** given in the SI (Figs. S9-S11).
2. The flake thickness is measured with VEECO profilometer, which is slightly unusual since AFM is more common tool. Hence the question about the sensitivity of this measurement technique.
3. An etching rate of ~10 nm per minute was measured and the dry etching time was modified accordingly for the different ~~substrate~~ *sample* thicknesses.
4. The resist mask is removed with only 3min of 50C treatment in acetone, which appears to be a very short time. Is there perhaps additional details confirming that all the resist is removed during this time? This can be quite important for the next patterning step of anisotropic etchant, since the top and side layers can modify the process.

Reply to Reviewer 1

This work reported a scalable method to fabricate hexagonally shaped TMD nanostructures with smooth edges, by combining beam-based nanolithography like EBL and FIB with anisotropic chemical etching. The demonstrated several applications of the technique. First, they demonstrated formation of ~ 3 nm wide WS₂ nanoribbons and nanojunctions, which are otherwise difficult to fabricate using existing fabrication techniques. Second they demonstrated honeycomb-array WS₂ metamaterials with tailored optical properties. The method is simple and could be extended to other 2D materials. The work is well executed with several innovations which should appeal to the 2D materials and metamaterials communities. However, the authors to not provide enough detail to reproduce the work, and the introduction is disorganized and does not do a good job motivating the work or addressing the most relevant previous works. After addressing these issues, I will recommend this paper for publication.

We are pleased to read a highly positive and constructive response of the reviewer. We believe that we can improve our manuscript by addressing the reviewers' questions and comments. Please find below a detailed point-by-point response to all the questions and concerns.

Comment 1. The authors did not provide nearly any details about the key innovation in their work: the anisotropic chemical etching process, which prevents the work from being reproduced. They need to clarify key etch parameters in the main text, such as the composition and concentration of the etchant, etch temperature, etch rate, and etch time, and any peripheral processing steps needed to make it work. They also need to describe how they choose the right chemical etchant.

Our reply: We thank the reviewer for this suggestion. Indeed, we agree that providing the details of our anisotropic etching process would be very helpful for the readers to reproduce the work and benefit for their future works. To address the comment, we have revised our manuscript and added a detailed Methods section at the end of the main text. This includes details regarding the descriptions of the key parameters including chemical concentration, temperature, time and other relevant conditions, STEM EELS data, optical spectroscopy and FDTD simulations.

In addition, more details about the composition of the etchant (H₂O₂/NH₄OH) were added in several relevant places throughout the entire manuscript.

Comment 2. The authors did not do a good job articulating what distinguishes their work compared to the most relevant examples literature. For instance, the paper *Advanced Materials* 2011, 23, 3061-3065 (Reference 24) demonstrated fabrication of nearly identical nanostructures in graphene by combining EBL with anisotropic plasma etch. Another paper *Nano Research* 2013, 6(3): 200-207 (Not referenced) reported anisotropic etching of MoS₂, generating

hexagonal patterns. This reviewer believes that this paper has novelty compared with those previous works, but it is important that the authors articulate how their work is distinct with a direct comparison to the two papers mentioned above, and to moderate their language about novelty of their own anisotropic etch process.

Our reply: We thank the reviewer for recognizing the novelty of our work. We agree that it is important to articulate the novelty of this work with respect to previous examples. We also thank the reviewer for bringing our attention to the Nano Research 2013 work, which is relevant and now cited.

In the revised manuscript we have done the following changes to address the reviewer's comment:

- We have explicitly discussed the relation between our work and most relevant literature examples in the introduction section (new references 24-26).
- We have rewritten the introduction, focusing more on etching details and comparison to previous literature. The reference to previous literature is extended (also thanks to other reviewer comments).

Comment 3. The discussion and structures in Figure 2 are difficult to follow. Figure 2D and the difference between monolayer and bilayer behavior is particularly difficult to interpret. This figure would benefit from cross-sectional diagrams of the proposed structure including the masking layers, so it is clear which layers are etching.

Our reply:

We thank the reviewer for this constructive comment. We would like to clarify that the difference between the mono- and bilayer WS_2 is shown specifically to address the thickness dependence of our method (all other panels in Fig. 2 report individual hole size, hole shape, and hole heterostructure study. The figure as a whole is devoted to individual holes and their parameters). For any thickness above the monolayer, we do not see significant variation in the final result. In fact, Figure S1 already shows a cross-sectional diagram of the fabrication process, which applies to all structures in this manuscript, including the structures shown in Fig. 2d.

The purpose of Fig. 2d is to show that under the same experimental conditions, monolayer and bilayers behave differently, that is, a bilayer can be patterned into hexagonal pattern, while a monolayer cannot. Fig. 2d also shows that the etching method does not operate along the basal plane and that excitons in WS_2 safely withstand the etching process, as is evident from the strong photoluminescence (PL) signal. We thus invested additional efforts to optimize etching monolayers. These details are discussed in Supplementary Notes (section d) and in Fig. S13. These observations are important hints towards understanding the etching mechanism, which we added into Discussion section.

Comment 4. The introduction is misleading and poorly organized. It spends a lot of time motivating general concepts of 2D materials, heterostructures and edge physics. This is misleading, because the paper does not demonstrate any property of edges or heterostructures. The introduction would be much improved by spending more time discussing the challenges and progress of etch control (including the papers mentioned above), then focusing on the progress and opportunities of the actual applications they focus on in the paper (e.g. metamaterials).

Our reply:

We thank the reviewer for this constructive comment. Following the reviewer's opinion, we have restructured the introduction (also taking into account Comment 2 of the reviewer), addressed above.

However, we would like to clarify that we have actually demonstrated several experimental observations related to zigzag edges, which include electron diffraction and EELS data in Fig. 1d-e, Fig. S6, as well as ultranarrow structures in Fig. 4. Specifically, ultranarrow nanoribbons and nanojunctions, are only possible to fabricate thanks to highly specific edges. The importance of zigzag edges is well motivated in the literature, however, the fabrication method of such structures for TMDs is lacking. That is why we feel that stressing edge physics and its relation to 2D materials in general in the introduction is important. To further stress this point, we have revised the manuscript in the Discussion section, where we have added several points related to fabrication of nanoribbons through formation of edges.

The metamaterial aspect in this work is somewhat broader, as we discuss the method of fabricating materials with combined edge-plane properties, instead of focusing the discussion specifically on metamaterials in the optical sense.

Reply to Reviewer 2

The authors report on anisotropic wet etching effect in transition metal dichalcogenide (TMD). The obtained edges in as-produced hexagonal holes are extremely smooth and exclusively zigzag-terminated. They further extend their study to fabricate more complex geometry pattern array and sub-10nm nanostructures including nanoribbons and nanojunctions. These results establish a facile and controllable method to tailor TMD materials, with a high potential in TMD-based electronics. The data are thoroughly presented. I would like to recommend its publication after major revisions.

We are very pleased to read a highly positive and constructive response of the reviewer.

Comment 1. A key point for this study is to use H₂O₂/NH₄OH solution replacing other reported etchants. However, the author described nothing about this issue in the text. I was totally puzzled until I checked supplementary materials. This is not advisable for a research article. The authors should provide a recipe regarding the composition and proportion. I would further inquire the etching mechanism, the role of NH₄OH and origination for the advantage of this etching solution as compared to other etchants.

Our reply: We thank the reviewer for this suggestion. Indeed, we agree that providing the details of our anisotropic etching process would be very helpful for the readers to reproduce the work and benefit for their future works. To address the comment, we have revised our manuscript and added a detailed Methods section at the end of the main text. This includes details regarding the descriptions of the key parameters including chemical concentration, temperature, time and other relevant conditions, STEM EELS data, optical spectroscopy and FDTD simulations.

In addition, more details about the composition of the etchant (H₂O₂/NH₄OH) were added in several relevant places throughout the entire manuscript.

A Discussion section focusing on the possible etching mechanism is added just before the conclusions. We have also expanded the description of various composition of the etchant (a mix of hydrogen peroxide/ammonia vs pure peroxide). The observation is that in presence of ammonia, the etching is faster (Figure S3).

A comparison of our etching process to other relevant examples is now provided in the Introduction section.

Comment 2. I doubt the claim of atomic precision. The author provided TEM data, showing edge roughness variation 1-2 atomic plane in the selected area with tens of nanometer in the length. Although it is not feasible to characterize the roughness in a large length scale as well as atomic precision, if the author could also provide roughness of 1-2 atomic plane for the triangle in the corner of hexagonal holes, the claim of atomic precision is then convincing.

Our reply: We thank the reviewer for this important comment. To address it we have added a corner region into revised Figure S4 (panel c), which shows a comparable level of sharpness. We also agree that it is hard to visualize atomic precision over extended length scales, however, the data presented in Figure 1e and Figure S4 is reproducibly obtained in different samples. We have revised the text in several places to highlight this point.

Comment 3. There is almost no quantitative description about the relationship between etching speed, etching time and etching condition, which is much important for the authors to design and fabricate nanostructures.

Our reply: We thank the reviewer for raising this point. In the revised manuscript we have provided more data, both in the main text and the SI. Specifically, Fig. S2 addresses the etching speed in a quantitative manner. Additionally, by addressing Comment 1, we included quantitative aspects (time, temperature, etching solution composition, additional details of EBL, RIE and FIB, etc.).

Comment 4. The authors mentioned the etching in monolayer is not anisotropic. Why? As I know, monolayer graphene on SiO₂ substrate also shows such anisotropic etching (Adv. Mater. 23, 3061 (2011)), but on substrate with atomically smooth surface like hBN shows anisotropic etching again (Appl. Phys. Lett. 109, 053103 (2016)). The authors need to provide detailed discussion on this issue.

Our reply:

We thank the reviewer for raising one of the most important questions in this study. Probably the reviewer means Appl. Phys. Lett. **109**, 053101 (2016), instead of Appl. Phys. Lett. **109**, 053103 (2016). Although the substrate effect is likely important, similarly to the case of graphene, we believe a more important aspect is that anisotropic etching of monolayers principally does not

lead to hexagons, instead they result in triangles. This point is shown in Figure S13 and covered thoroughly in Figure 2 and the new Discussion section.

The revised details of an optimal conditions for anisotropic etching in monolayers is discussed in the Supplementary Notes (section d).

Reply to Reviewer 3

The article “Transition metal dichalcogenide metamaterials with atomic precision” presents a fabrication technique to produce nanopatterned structures from WS₂ layers that can be further applied to a wider family of TMD materials, such as MoS₂ and MoSe₂. Moreover, the authors suggest that the developed fabrication technique can result in atomically sharp edges such as zigzag terminated structures. Authors present successful fabrication using this technique of a wide range of interesting nanostructures, including particularly interesting results on heterostructures. In general, the control over the TMD material edge structure is of high importance for a wide range of applications, since such nanoscale precision patterning techniques give an opportunity to not only control electronic and optical properties of the materials but also create novel properties such as for example ferromagnetism. Therefore, the presented results are of high importance and interest to the wide reader audience of the Nature Communication journal. However, the current manuscript lacks important discussions such mechanism of chemical treatment, material quality preservation and photonic structure properties. Please see the major and minor correction list below. At this stage, the manuscript is not ready for publication, but it can be improved after the major corrections being addressed.

We are pleased to read a highly positive and constructive response of the reviewer.

To address the comment, we have revised our manuscript and added a detailed Methods section at the end of the main text. This includes details regarding the descriptions of the key parameters including chemical concentration, temperature, time and other relevant conditions, STEM EELS data, optical spectroscopy and FDTD simulations.

In addition, more details about the composition of the etchant (H₂O₂/NH₄OH) were added in several relevant places throughout the entire manuscript.

Comment 1. WS₂ edge termination: The authors suggest that after the fabrication technique, the WS₂ structure termination has atomically sharp zigzag nature. However, in fig.1(e) it is very difficult to see the actual edge. The authors are strongly advised to include higher magnification image and atomic model of the material structure to justify the edge termination. It would be also very interesting to see the crystal corner similarly to the work by *Chen, Q. et al. Atomically Flat Zigzag Edges in Monolayer MoS₂ by Thermal Annealing. Nano Lett. 17, 5502-5507, (2017).* In this work, Chen et al has also demonstrated that there are atomic corrugations present on the formed edge produced by thermal annealing, it would be very interesting to compare this result to the results of the proposed method.

Our reply: We thank the reviewer for constructive comment. We agree that it is hard to see the zigzag edge character directly from the TEM images. This is partially due to the multilayer nature of our samples, which are harder to visualize than a monolayer. For this reason, in our

assignment of edge termination we rely mostly on the electron diffraction, which unambiguously proves these edges are exclusively aligned with the zigzag termination direction (see Fig. 1d). In addition to that, EELS data shown in Figure S6, clearly shows a peak at 13.8 eV, which was previously attributed to zigzag edges. To obtain a high resolution TEM image of edge termination, an experiment focusing on a monolayer, preferably above vacuum, is required. This, however, goes beyond this study.

Following, the reviewer's suggestion we have added a TEM image of the corner area in Figure S4 (panel c). This image shows a similar surface roughness of the crystalline material near the corner areas.

Comment 2. Wet etching process: The presented fabrication technique relies fully on the anisotropic etching process activated by H₂O₂. However, the manuscript doesn't include any details on the etchant composition and the authors just refer to the solution as "etchant", which is quite uninformative. Furthermore, there is no description of the mechanism of this etchant anisotropic functions. The authors are therefore strongly advised to include the chemical mechanism description. It might be very interesting to look into this, since it might further explain the results presented for single layer WS₂ where triangular structures are formed. This can be a very nice discussion section where all the results for various thicknesses and shapes are summarised and explained. Furthermore, it is also quite interesting why this etching technique works for MoS₂ and MoSe₂, but fails for WSe₂. Additionally, was the crystal thickness same for the tested structures of different materials? Based on the results presented for various shapes in Fig2, it's quite interesting to see when the self-termination happens, perhaps it might be possible to develop a model describing the self termination process similar to results presented in Sang, X. et al. In situ edge engineering in two dimensional transition metal dichalcogenides. Nat. Commun. 9, 2051, (2018).

Our reply:

We thank the reviewer. This comment was raised by all the reviewers. To address the comment, we have revised our manuscript and added a detailed Methods section at the end of the main text. This includes details regarding the descriptions of the key parameters including chemical concentration, temperature, time and other relevant conditions, STEM EELS data, optical spectroscopy and FDTD simulations.

In addition, more details about the composition of the etchant (H₂O₂/NH₄OH) were added in several relevant places throughout the entire manuscript.

We also followed the reviewer's suggestion about including a discussion about the mechanism of anisotropic etching and relating between mono- and multilayer etching. We agree that it is highly

interesting. A comparison of multi- vs monolayer etching and following from that hints about the mechanism of anisotropic etching is now included in the Discussion chapter. However, we also feel that future studies will be needed to provide details of etching mechanism.

We also hypothesize that WSe₂ is likely more reactive to hydrogen peroxide. This is partially confirmed by a previous study showing self-limiting etching by ozone *Nano Letters* **15**, 2067-2073, (2015). All tested materials MoS₂, MoSe₂, WS₂, WSe₂ had similar flake thicknesses.

We also agree that it is very interesting to know where the self-termination happens. In fact, our data presented for monolayers (Figure 2d and Figure S13) and multilayers (the rest of the study), suggests that self-termination for monolayers and multilayers is different. In monolayers, it is very often a “saw-teeth” like patterns, clearly visible in Figure S13, while for multilayers, it is always hexagons. Based on this we have proposed the mechanism of anisotropic etching in the revised Discussion section. However, a more detailed self-termination model, like the one presented by Sang X *et al.*, would require significant extra effort and goes beyond the current study.

Comment 3. Material quality preservation: Another very important question is how much damage is introduced during the etching process. While the PL spectrum shown in the Fig. 2d shows high quality WS₂ in the middle, it would be very interesting to see the line scan of PL or Raman for example to estimate how much material is damaged close to the edge of the structures. This becomes a very crucial question especially for a nanojunction structures presented in Fig. 4. Additionally, it would be very nice to see the PL comparison for the material before and after patterning, since this can give additional measure of the induced defects, although the excitonic states will probably be modified too. From this point of view, having PL measurements for structured materials would be potentially quite interesting to study.

Our reply: We agree with the reviewer that material properties preservation is a crucial characteristic of the etching method. The PL data indeed has a diffraction limited spatial resolution and thus is hard to judge on the quality of edges. For this reason, we used EELS with a sub nanometer resolution, the result of which in Fig. 1e (inset) and Fig. S5 clearly demonstrates A-, B-, and C- exciton preservation right at the edge. If excitons would be damaged at the edge, we would expect a completely different picture.

Comment 4. Photonic structure properties: The presented fabrication technique results in very interesting photonic structures as presented in Fig.3. It would be very nice to see the actual measured reflectance spectra for unstructured and structured material regions, as well as supporting substrate. Additionally, the reader would definitely benefit from small SEM images of each array to be able to quickly judge how much WS₂ material is actually left after patterning.

For example, array N7 is claimed to have high absorption, however there is probably quite little WS₂ material left at least based on the structure sizes mentioned. Perhaps the observed reflectance results can be supported by numerical calculations, such as FDTD calculations, which should be quite reasonable for 70nm thick WS₂. While Fig.3a,b are quite nice demonstrations of the fabrication technique cases, it appears that these results can be either merged with Fig.2 or moved to the supplementary. The reader would probably benefit more from a more detailed figure on the photonic crystals. The paper might be stronger with an additional deeper discussion of the expected photonic structure properties, i.e. a theoretical part with calculations of the expected resonant features for the selected photonic crystal geometries. This becomes particularly relevant since the article title is *metamaterials*.

Our reply:

We thank the reviewer for this extended and constructive comment. To address the comment, we have modified Fig. 3 and included SEM data of the honeycomb arrays, as well as FDTD simulations. The reviewer is right that array #7 has the least amount of WS₂, however, additional absorption happens in the underlying Si substrate. This is now evidenced by FDTD in modified Figure 3 and new Figure S18.

In the revised text, we have also clarified that, although circular holes would produce essentially the same photonic modes, its combination of zigzag edges may open possibilities for cross-fertilization between applications of zigzag edges and nanophotonics.

The modes of photonic crystals are also discussed now in more detail. A comparison between experiment and theory is now provided. We would just like to clarify here that the metamaterial aspect of this work goes beyond optical sense. We rather focus on multifunctional materials, which combine characteristics of photonics, edges, excitons, and electronics in one single package.

Comment 5. Nanoribbons: Fig. 4 shows very interesting results demonstrating nanoribbons and nanojunctions. However, all the presented measurements are done with SEM, which is known to locally modify WS₂ as well. AFM measurements would give more precise results on the nanostructure sizes, while not modifying the 2D materials. Additionally, Fig. 4a clearly shows an internal ring structure that is not discussed in the text, are those residues or there is an etching tilt or this is etching in SiO₂? Similar rings are observed in Fig.2 d and c.

Our reply: We thank the reviewer for the comment and noticing important details. Indeed, we are aware of that SEM modify the excitonic properties locally. However, we would like to emphasize that we use SEM to simply visualize the fine details of the vertical zigzag edges. Otherwise, SEM is not a part of our fabrication process, and thus can be fully omitted in case

preservation of excitonic (and other material properties) is of concern. We also appreciate the suggestion to use AFM measurements, which we unfortunately could not do in this study, but it will certainly be used in future work.

We also thank the reviewer for noticing the fine detail in Fig. 4a. The internal circular structures in Fig. 4a as well as in Fig. 2 are indeed etched circular holes in SiO₂/Si substrate below the TMD flakes, formed during the RIE etching. More importantly, it reveals the initial design of circular hole before our anisotropic etching process. One can also see that the quality and limitations of traditional EBL and RIE techniques for nanopatterning TMDs. In contrast, we believe that our anisotropic etching method can overcome imperfections of the tradition techniques and “heal” the defects created by EBL/RIE to result in perfect hexagons with exclusive zigzag edges. This makes our method unique. The discussion of this point is now added into revised manuscript.

Comment 6. FIB results are quite interesting especially from the point of view of polymer residue influence on the process. However, they are not discussed in the main text.

Our reply: We thank the reviewer for suggestion expanding our discussions on the FIB results. Indeed, we fully agree with the reviewer from the point of view of using maskless-lithography techniques, e.g. FIB or laser patterning, to prevent potential polymer residue effects on the process. This is mentioned in the generic fabrication scheme in Figure S1. We successfully combined FIB ion milling technique with our anisotropic etching method. The result is summarized in Fig. S17. However, we noticed that excitonic properties of TMD materials is modified substantially during the FIB session (for which reason this data is not a part of the main text). It could be attributed to Ga⁺ ion implantation, amorphous contamination, and degradation due to SEM imaging during the FIB session. These problems may be potentially overcome by employing FIB imaging (instead of SEM) as well as by using alternative ions in FIB, for instance, helium. Our anisotropic technique can be easily combined with various types of milling or traditional patterning techniques.

Comment 7. Additional data about the quality of single holes, more complicated structures such as bull’s eye nanostructures, as well as tests performed on multiple semiconductor group VI TMD materials ~~is~~ are given in the SI (Figs. S9-S11).

Our reply: We thank the reviewer for pointing out this error. We have replaced “is” with “are”.

Comment 8. The flake thickness is measured with VEECO profilometer, which is slightly unusual since AFM is more common tool. Hence the question about the sensitivity of this measurement technique.

Our reply: We thank the reviewer for the comment. In our experiment, we used Dektak 150 profilometer in the clean room environment, which provides accurate and repeatable measurements on various surfaces in convenient manner, from traditional 2D roughness surface characterization and step-height measurements to advanced 3D mapping and film stress analysis. Dektak 150 enables superior performance and versatility with step-height repeatability/resolution of 0.6 nm as well as vertical resolution of 0.1 nm. Thus, we believe that Dektak-150 profilometer is an accurate instrument to determine thicknesses of multilayer TMDs. In addition, mono- and bilayer TMDs can be identified via direct PL measurements.

Comment 9. An etching rate of ~10 nm per minute was measured and the dry etching time was modified accordingly for the different ~~substrate~~ *sample* thicknesses.

Our reply: We thank the reviewer for pointing out this error. We have replaced “substrate” with “sample”.

Comment 10. The resist mask is removed with only 3min of 50C treatment in acetone, which appears to be a very short time. Is there perhaps additional details confirming that all the resist is removed during this time? This can be quite important for the next patterning step of anisotropic etchant, since the top and side layers can modify the process.

Our reply: We thank the reviewer for noticing another important detail. During the reactive ion etching (RIE), not only TMD, but also the resist mask is also etched, that leads to relatively thin layer of the resists. However, during the RIE etching, the resist mask is known to be hardened. Therefore, our etching recipe includes also oxygen plasma stripping right after RIE etching to remove not only hardened top layer of resists, but also substantial amount of the resist. Thus, thickness of the resists is reduced substantially during the dry etching process. Therefore, left-over resist can be removed within relatively short time in Acetone. To address reviewer's comment, we have updated the Methods section with additional discussion.

Moreover, even if a small amount of organic contaminants is left on the surface of TMD after the RIE step, it will be additionally cleaned out by hydrogen peroxide due to its known ability to oxidize organic contaminants.

We have also tested soaking the sample in Acetone overnight to remove residual resists and performed our anisotropic etching method. We did not observe any differences in the quality of hexagons between overnight soaking and short removal of resists.

REVIEWERS' COMMENTS:

Reviewer #1 (Remarks to the Author):

The authors have addressed this reviewers major comments, and this reviewer believes the manuscript is ready for publication after the authors consider one minor comment:

As per comment 3 from my previous review. The authors explained what was happening in Figure 2, which was never the issue in the comment. The comment was on the presentation of the figure. The main text Figure 2 would benefit strongly from some cartoon profile schematics (not data/images of profiles, and not in the supporting information) to help readers interpret the images as the top down images and fine lines make it difficult to understand what is happening in each layer during the masked etch process. This comment is minor, and should not hold up publication with another round of review, but I would strongly suggest that the authors add these schematics.

Reviewer #2 (Remarks to the Author):

Authors have addressed all my concerns in their report and revised the manuscript for a better version, I thus recommend its publication in NC.

Reply to Reviewer 1

The authors have addressed this reviewers major comments, and this reviewer believes the manuscript is ready for publication after the authors consider one minor comment:

As per comment 3 from my previous review. The authors explained what was happening in Figure 2, which was never the issue in the comment. The comment was on the presentation of the figure. The main text Figure 2 would benefit strongly from some cartoon profile schematics (not data/images of profiles, and not in the supporting information) to help readers interpret the images as the top down images and fine lines make it difficult to understand what is happening in each layer during the masked etch process. This comment is minor, and should not hold up publication with another round of review, but I would strongly suggest that the authors add these schematics.

Our reply: We thank the reviewer for positive evaluation of our manuscript.

We also thank the reviewer for the comment on Fig. 2. To address the comment, we have modified Fig. 2 and included a schematic sketch.

Reply to Reviewer 2

Authors have addressed all my concerns in their report and revised the manuscript for a better version, I thus recommend its publication in NC.

Our reply: We thank the reviewer for positive evaluation of our manuscript.